# Transcriptome and Proteomic Analysis Reveals Up-Regulation of Innate Immunity-Related Genes Expression in Caprine Herpesvirus 1 Infected Madin Darby Bovine Kidney Cells

**DOI:** 10.3390/v13071293

**Published:** 2021-07-02

**Authors:** Fei Hao, Xing Xie, Maojun Liu, Li Mao, Wenliang Li, Woonsung Na

**Affiliations:** 1Key Laboratory for Veterinary Bio-Product Engineering, Ministry of Agriculture and Rural Affairs, Institute of Veterinary Medicine, Jiangsu Academy of Agricultural Sciences, Nanjing 210014, China; 20140010@jaas.ac.cn (F.H.); 20160042@jaas.ac.cn (X.X.); 20030022@jaas.ac.cn (M.L.); 20100014@jaas.ac.cn (L.M.); 2School of Food and Biological Engineering, Jiangsu University, Zhenjiang 212013, China; 3College of Veterinary Medicine, Chonnam National University, Gwangju 61186, Korea; 4Animal Medical Institute, Chonnam National University, Gwangju 61186, Korea

**Keywords:** caprine herpesvirus 1, innate immune response, ISGs, RNA-Seq, iTRAQ

## Abstract

Caprine herpesvirus 1 (CpHV-1) is a member of the alpha subfamily of herpesviruses, which is responsible for genital lesions and latent infections in goat populations worldwide. In this study, for the first time, the transcriptome and proteomics of CpHV-1 infected Madin Darby bovine kidney (MDBK) cells were explored using RNA-Sequencing (RNA-Seq) and isobaric tags for relative and absolute quantitation-liquid chromatography tandem mass spectrometry (iTRAQ-LC-MS/MS) technology, respectively. RNA-Seq analysis revealed 81 up-regulated and 19 down-regulated differentially expressed genes (DEGs) between infected and mock-infected MDBK cells. Bioinformatics analysis revealed that most of these DEGs were mainly involved in the innate immune response, especially the interferon stimulated genes (ISGs). Gene Ontology (GO) enrichment analysis results indicated that the identified DEGs were significantly mainly enriched for response to virus, defense response to virus, response to biotic stimulus and regulation of innate immune response. Viral carcinogenesis, the RIG-I-like receptor signaling pathway, the cytosolic DNA-sensing pathway and pathways associated with several viral infections were found to be significantly enriched in the Kyoto Encyclopedia of Genes and Genomes (KEGG) pathway database. Eleven selected DEGs (Mx1, RSAD2, IFIT1, IFIT2, IFIT5, IFIH1, IFITM3, IRF7, IRF9, OAS1X and OAS1Y) associated with immune responses were selected, and they exhibited a concordant direction both in RNA-Seq and quantitative real-time RT-PCR analysis. Proteomic analysis also showed significant up-regulation of innate immunity-related proteins. GO analysis showed that the differentially expressed proteins were mostly enriched in defense response and response to virus, and the pathways associated with viral infection were enriched under KEGG analysis. Protein-protein interaction network analysis indicated most of the DEGs related to innate immune responses, as DDX58(RIG-I), IFIH1(MDA5), IRF7, Mx1, RSAD2, OAS1 and IFIT1, were located in the core of the network and highly connected with other DGEs. Our findings support the notion that CpHV-1 infection induced the transcription and protein expression alterations of a series of genes related to host innate immune response, which helps to elucidate the resistance of host cells to viral infection and to clarify the pathogenesis of CpHV-1.

## 1. Introduction

Herpesviruses are double-stranded DNA enveloped viruses and are regarded as important human and animal pathogens. They heavily impact human health under a number of conditions and cause economic losses in livestock [1,2]. Caprine herpesvirus 1 (CpHV-1) belongs to the family *Herpesviridae* and the genus *Varicellovirus*. The virus was first isolated from goats in 1974 and further characterized in 1975 [3]. CpHV-1 infection has been reported in goats globally, including in the US, Canada, Brazil, Australia, New Zealand, Mediterranean countries (such as Greece, Italy, Spain, France, etc.) and China [4,5,6,7,8]. CpHV-1 is believed to infect goats by the nasal or reproductive route, displaying high tropism to the genital tract [9,10]. The virus can establish latent infection in trigeminal ganglia and eventually induce immunosuppression. CpHV-1 causes systemic disease in young kids, characterized by high morbidity and mortality rates, while in adult goats the infection leads to vulvovaginitis, balanoposthitis, respiratory disease and occasionally abortion [6,7,11,12,13]. Previous studies showed that CpHV-1 could induce apoptosis in goat peripheral blood mononuclear cells and the Madin Darby bovine kidney (MDBK)cell line [14,15]. Emerging evidence has indicated that CpHV-1 could infect varied human cell lines and provide a potential candidate for oncolytic virotherapy [16]. Moreover, it was reported that CpHV-1 replicated efficiently in experimentally infected calves during acute infection and established latent infection, which supports the ability of the virus to cross-infect the respective heterologous hosts [17,18]. However, the extent and relevance of CpHV-1-mediated host responses and the relationship between this responses and viral pathogenicity remain poorly studied.

Recently, large-scale screening has been recognized to be very effective for identifying host genes and proteins of interest upon viral infection, many of which are unexpected and may lead to new discoveries regarding the host–virus interactions after additional functional validation. Transcriptome and proteomic analysis provide a thorough understanding of the host defense mechanisms and immune evasion strategies of viral infection. Both high-throughput RNA-Sequencing (RNA-Seq) technologies and isobaric tags for relative and absolute quantitation (iTRAQ), in combination with liquid chromatography tandem mass spectrometry (LC-MS/MS) analysis, have become powerful methods to explore pathogen–host interactions. In this study, Illumina sequencing method was used to identify the transcriptome changes in CpHV-1 infected MDBK cells. For the first time, the differentially expressed transcriptome profile in the MDBK cells was obtained during CpHV-1 infection. At the same time, iTRAQ was used to identify the differentially expressed proteins (DEPs) in MDBK cells infected with CpHV-1. These data provide clues to further understanding the replication and pathogenesis of CpHV-1 and virus–host interactions.

## 2. Materials and Methods

### 2.1. Cell Culture, Virus Infection and Sample Preparation

Madin Darby bovine kidney (MDBK) cells were purchased from the China Institute of Veterinary Drug Control. The cells were grown in Dulbecco’s Modified Eagle Media (DMEM, Hyclone, Logan, UT, USA), supplemented with 10% fetal bovine serum (FBS, Transgen, Beijing, China) at 37 °C with 5% CO_2_. MDBK cells were suspended in 1 × 10^6^ cells/mL with DMEM containing 10% FBS and seeded in six-well plates (*n* = 4, 6 wells/sample). Two plates were inoculated with the CpHV-1 strain JSHA1405, with the GenBank accession number MG989243 (10^8^ TCID_50_/mL) [8], at a multiplicity of infection (MOI) of 1. The other two plates were used as mock-infected control. The CpHV-1-infected and mock-infected cells were harvested at 12 h and 24 h post-infection (hpi) through scraping, followed by centrifugation at 5000× *g* for 10 min at 4 °C. Each cell sample (pooled of the cells in one plate) was washed twice with ice-cold phosphate buffered saline (PBS) and subjected to the following experiment. Samples harvested at 12 hours post infection (hpi) were mixed and pelleted for RNA-Seq, and samples harvested at 24 hpi were mixed and pelleted for iTRAQ technology.

### 2.2. RNA Extraction, Library Preparation and RNA-Seq

As previously described [19], total RNA was extracted from the CpHV-1-infected (C-1) and mock-infected (N) MDBK cells using TRIzol^®^ Reagent (Invitrogen, Carlsbad, CA, USA) according the manufacturer’s instructions, and genomic DNA was removed using DNase I (TaKara, Dalian, Liaoning, China). Then, RNA quality was determined using 2100 Bioanalyser (Agilent, Santa Clara, CA, USA) and quantified using the ND-2000 (NanoDrop Technologies, Wilmington, DE, USA). A high-quality RNA sample (OD260/280 = 1.8~2.2, OD260/230 ≥ 2.0, RIN ≥ 6.5, 28S:18S ≥ 1.0, >10 μg) was used to construct the sequencing library. 

RNA-seq transcriptome libraries were prepared using the TruSeq RNA Library Preparation Kit (Cat No. RS-122-2001, Illumina, San Diego, CA, USA) and 5μg of total RNA. Messenger RNA was isolated with polyA selection by oligo(dT) beads and fragmented using fragmentation buffer. cDNA synthesis, end repair, A-base addition and ligation of the Illumina-indexed adaptors were performed according to Illumina’s protocol. Libraries were then size-selected for cDNA target fragments of 200–300 bp on 2% Low Range Ultra Agarose followed by PCR amplified using Phusion DNA polymerase (NEB, Ipswich, MA, USA) for 15 PCR cycles. After being quantified by TBS380, paired-end libraries prepared for both the CpHV-1-infected and mock-infected samples (C-1 and N) were sequenced by Illumina NovaSeq 6000 sequencing (Illumina, San Diego, CA, USA, 2 × 150 bp read length, Shanghai BIOZERON Co., Ltd.).

### 2.3. Read Quality Control and Mapping

The raw paired-end reads were trimmed and quality controlled by Trimmomatic with parameters (SLIDINGWINDOW:4:15 MINLEN:75) (version 0.36 http://www.usadellab.org/cms/uploads/supplementary/Trimmomatic, accessed on 7 March 2019). Clean reads were then separately aligned to the reference bovine genome (https://www.ncbi.nlm.nih.gov/assembly/GCF_002263795.1, accessed on 21 February 2019), with orientation mode using hisat2 (https://ccb.jhu.edu/software/hisat2/index.shtml, accessed on 25 February 2019) software, which was used to map with default parameters. The quality assessment of these data was conducted using qualimap_v2.2.1(http://qualimap.bioinfo.cipf.es/, accessed on 11 December 2018) and htseq (https://htseq.readthedocs.io/en/release_0.11.1/, accessed on 4 February 2019) to count each gene read. 

### 2.4. Differential Expression Genes (DEGs) Analysis, Functional Enrichment, and Protein–Protein Interaction (PPI) Analysis of DEGs

To identify the DEGs between the two different samples, the expression levels for each gene were calculated using the fragments per kilobase of exon per million mapped reads (FRKM) method. R statistical package edgeR (Empirical analysis of Digital Gene Expression in R, http://www.bioconductor.org/packages/release/bioc/html/edgeR.html/, accessed on 12 March 2019) was used for differential expression analysis. The DEGs between the two samples were selected using the following criteria: the logarithmic of fold change was greater than 2, and the false discovery rate (FDR) was less than 0.05. To understand the functions of differentially expressed genes, Gene Ontology (GO) functional enrichment and Kyoto Encyclopedia of Genes and Genomes (KEGG) pathway analysis were carried out by Goatools (https://github.com/tanghaibao/Goatools, accessed on 15 March 2019) and KOBAS (http://kobas.cbi.pku.edu.cn/kobas3, accessed on 15 March 2019). DEGs were significantly enriched in GO terms and metabolic pathways when their Bonferroni-corrected *p*-value was less than 0.05. The PPI network among the DEGs was analyzed using the STRING (http://string-db.org/, accessed on 2 March 2019) database, which included direct and indirect associations of proteins. After analyzing the result from STRING analysis and expression change information for each DEG, the network figure was drawn for the selected DEGs (connected with one or more DEGs) using Cytoscape software (https://cytoscape.org/, accessed on 23 January 2019) [19].

### 2.5. Quantitative Real-Time RT-PCR (qRT-PCR)

To validate the DEGs revealed by the transcriptome data, leaves of CpHV-1-infected (C-1) and mock-infected (N) MDBK cells were collected. Total RNA was extracted from C-1 and N using TRIzol^®^ Reagent (Invitrogen, Carlsbad, CA, USA) according the manufacturer’s instructions. The qRT-PCR amplification was carried out with TransScript one-step qRT-PCR Supermix (Transgen, Beijing, China) in a 20 μL reaction mixture containing 10 μL of 2 × Supermix, 20 pM of each primer (Table 1), 0.5 μL of E-Mix, 0.4 μL of passive reference Dye and 2 μL of extracted RNA. The reaction was run on a StepOnePlus Real-Time PCR System (Applied Biosystems, Carlsbad, CA, USA) with the following procedure: samples were incubated at 45 °C for 5 min before being heated at 94 °C for 30 s, and a two-step cycle (5 s at 94 °C, 30 s at 60 °C) was repeated for 40 cycles. β-actin was used as the internal control, and relative quantification of target gene expression was used to compare the target transcript in the infected group to that of mock-infected group; the data analysis was based on 2-ΔΔCt method.

### 2.6. Total Protein Extraction

The sample was ground individually in liquid nitrogen and lysed with lysis buffer containing 100 mM NH_4_HCO_3_ (pH 8.0), 8 M Urea and 0.2% SDS, followed by 5 min of ultrasonication on ice. The lysate was centrifuged at 12,000× *g* for 15 min at 4 °C, and the supernatant was transferred to a clean tube. Extracts from each sample were reduced with 10 mM Dithiothreitol (DTT) for 1 h at 56 °C and subsequently alkylated with sufficient iodoacetamide for 1 h at room temperature in the dark. Then, samples were completely mixed with 4 times the volume of precooled acetone by vortexing and incubated at −20 °C for at least 2 h. Samples were then centrifuged, and the precipitation was collected. After washing twice with cold acetone, the pellet was dissolved by dissolution buffer, which contained 0.1 M triethylammonium bicarbonate (TEAB, pH 8.5) and 6 M Urea.

### 2.7. Protein Quality Test

Bovine serum albumin (BSA) standard protein solution was prepared according to the instructions of the Bradford protein quantitative kit, with gradient concentration ranging from 0 to 0.5 g/L. BSA standard protein solutions and sample solutions with different dilution multiples were added into 96-well plate to fill up the volume to 20 μL. Each gradient was repeated three times. To the plate was added 180 μL G250 dye solution quickly, and it was placed at room temperature for 5 min; the absorbance at 595 nm was detected. The standard curve was drawn with the absorbance of standard protein solution, and the protein concentration of the sample was calculated. A total of 20 μg of the protein sample was loaded to 12% SDS-PAGE gel electrophoresis, wherein the concentrated gel was performed at 80 V for 20 min, and the separation gel was performed at 120 V for 90 min. The gel was stained by coomassie brilliant blue R-250 and decolored until the bands were visualized clearly.

### 2.8. iTRAQ Labeling of Peptides

Each sample of about 100 μg was labeled using the iTRAQ Reagent-8Plex Multiplex kit (Cat No. 4390812, AB Sciex, Framingham, MA, USA) in accordance with the manufacturer’s instructions. After, in order to check the iTRAQ labeling efficiency, one third of samples mentioned were mixed and prepared for Liquid chromatography-electrospray ionization tandem MS (LC-MS/MS) mass spectrometry detection.

### 2.9. Separation of Fractions

Mobile phase A (2% acetonitrile, adjusted pH to 10.0 using ammonium hydroxide) and B (98% acetonitrile) were used to develop a gradient elution. The lyophilized powder was dissolved in solution A and centrifuged at 14,000× *g* for 20 min at 4 °C. The sample was fractionated using a C18 column (Waters BEH C18 4.6 × 250 mm, 5 μm) on an HPLC system, and the column oven was set as 50 °C. The eluates were monitored at UV 214 nm, collected for a tube per minute and combined into 10 fractions finally. All fractions were dried under vacuum and then reconstituted in 0.1% (*v*/*v*) formic acid (FA) in water.

### 2.10. LC-MS/MS Analysis

For transition library construction, shotgun proteomics analyses were performed using an EASY-nLCTM1200 UHPLC system (Thermo Fisher, Rockford, IL, USA) coupled with an Q Exactive HF (X) mass spectrometer (Thermo Fisher, Rockford, IL, USA) operating in the data-dependent acquisition (DDA) mode. The sample was injected into a homemade C18 Nano-Trap column (2 cm × 100 μm, 3 μm). 

Peptides were separated in a homemade analytical column (12 cm × 150 μm, 1.9 μm). The separated peptides were analyzed by Q Exactive HF (X) mass spectrometer (Thermo Fisher, Rockford, IL, USA), with ion source of Nanospray Flex™ (ESI), spray voltage of 2.5 kV and ion transport capillary temperature of 320 °C. The full scan range was from *m*/*z* 407 to 1500 with resolution of 60,000 (at *m*/*z* 200), the automatic gain control (AGC) target value was 3 × 10^6^ and the maximum ion injection time was 20 ms. The top 40 precursors of the highest abundance in the full scan were selected and fragmented by higher energy collisional dissociation (HCD) and analyzed in MS/MS [20], where the resolution was 15,000 (at *m*/*z* 200), the automatic gain control (AGC) target value was 5 × 10^4^, the maximum ion injection time was 45 ms, the normalized collision energy was set as 32%, the intensity threshold was 2.2 × 10^4^ and the dynamic exclusion parameter was 20 s.

### 2.11. The Identification and Quantitation of Protein

The resulting spectra from each run were searched separately against Enzyme database by the search engine Proteome Discoverer 2.2 (PD 2.2, Thermo Fisher, Rockford, IL, USA). The searched parameters were set as follows: mass tolerance for precursor ion was 10 ppm and mass tolerance for product ion was 0.02 Da. Carbamidomethyl was specified as fixed modifications, Oxidation of methionine (M) and iTRAQ plex were specified as dynamic modification, acetylation and iTRAQ plex were specified as N-Terminal modification in PD 2.2. A maximum of 2 miscleavage sites were allowed. 

In order to improve the quality of the analysis results, the software PD 2.2 further filtered the retrieval results; Peptide Spectrum Matches (PSMs) with a credibility of more than 99% were identified as PSMs. The identified protein contained at least 1 unique peptide. The identified PSMs and protein were retained and performed with an FDR of no more than 1.0%. The protein quantitation results were statistically analyzed by T-test. The proteins whose quantitation significantly different between the CpHV-1-infected group (C-1) and mock-infected control groups were defined as differentially expressed proteins (DEPs) (fold change (FC) ≥ 1.2 or ≤ 0.83).

### 2.12. The Functional Analysis of Protein

Clusters of Orthologous Groups (COG), GO and KEGG [21] were used to analyze the protein family and pathway. DEPs were used for Volcanic map analysis, cluster heat map analysis and enrichment analysis of GO and KEGG.

### 2.13. Western Blot

Cell lysates samples were separated by 12% SDS-PAGE and transferred onto PVDF membrane (Millipore, Darmstadt, Germany) using a semi-dry transfer cell (Bio-Rad, Hercules, CA, USA) at 1 V/cm^2^ for 40 min. The membrane was treated sequentially with 5% skimmed milk in TBS containing 0.5% Tween 20 (TBST) at 37 °C for 2 h, with different primary antibodies (1/200 diluted rabbit anti-Viperin polyclonal antibody (Abcam, Cambridge, UK), 1/200 diluted mouse anti-ISG15 antibody (Santa Cruz, Dallas, TX, USA, provided by Pfo. Bin Zhou, Nanjing agricultural university), 1/200 diluted mouse anti-MX1 antibody (Santa Cruz, Dallas, TX, USA, provided by Pfo. Bin Zhou, Nanjing agricultural university), 1/5000 diluted anti-β-actin monoclonal antibody (Transgen, Bio, Inc., Beijing, China) at 37 °C for 2 h, and with different secondary antibodies (1/1000 diluted rabbit anti-mouse or goat anti-rabbit IgG antibody conjugated to HRP (Transgen, Bio, Inc., Beijing, China). After three washes with TBST, the color development was performed using enhanced chemiluminescence luminal reagent (Thermo Scientific Pierce, Rockford, IL, USA).

### 2.14. Statistical Analysis

All experiments were reproducible and carried out in triplicate. The integrated density of every group performed by western blotting was compared with that of internal loading control β-actin, before the relative integrated density values between every infected group and its corresponding control group were calculated. Paired *t* test was used to determine statistical significance. The differences in relative integrated density of viperin, ISG15 and Mx1 between virus-infected cell and control cell groups were assessed by GraphPad Prism 6 software. Differences were considered statistically significant at a *p* value of 0.05, and *p* < 0.01 was extremely significant.

## 3. Results

### 3.1. Transcriptome Quantification

After RNA-seq, a total of 56,289,348 raw reads (C-1: 30,065,892; N: 26,223,456) were obtained. After removing low-quality reads and reads with adaptor sequences, 48,255,018 clean reads (C-1: 25,480,936; N: 22,774,082) were obtained (Table 2). We then queried the clean reads against the latest reference genome (ARS-UCD1.2, https://www.ncbi.nlm.nih.gov/assembly/GCF_002263795.1, accessed on 15 January 2019) and mapped using TopHat (http://tophat.cbcb.umd.edu/, accessed on 15 January 2019). For C-1 and N samples, 19,704,426 and 17,675,904 reads were mapped to the reference genome, with a mapped rate of 77.33% and 77.61%, respectively. Among the matched 13,345 target genes for C-1, 10,888 had reads per kilobase million (RPKM (FPKM)) values > 1; For N, 13,153 target genes were matched, and 10,864 had RPKM > 1 (Table 2).

### 3.2. DEG Analysis and Functional Annotation

After the gene mapping and the Cuffdiff analyses in terms of FPKM, a total of 100 genes were identified as significantly differentially expressed for the CpHV-1-infected group (C-1), when compared with the mock-infected control group (N, fold change (FC) ≥ ± 2, *p* < 0.05). Among the 100 genes, 81 were up-regulated and 19 were down-regulated (Figure 1 and Appendix A). DEGs related to innate immunity, such as IFIH1(MDA5), IRF7, IRF9, IFIT1/2, Mx1, RSAD2, OAS, etc., were found to be significantly up-regulated (Table 3 and Appendix A).

The 100 DEGs were annotated to 38 different GO terms. The up-regulated DEGs were annotated to 38 GO terms, and the down-regulated DEGs were annotated to 22 GO terms (Figure 2). The most annotated GO terms were cellular process (biological process, BP), biological regulation (BP), response to stimulus (BP), cell (cellular component, CC), cell part (CC), organelle (CC) and binding (molecular function, MF) (Figure 2).

### 3.3. Functional and PPI Analysis of DEGs

DEGs were enriched into different GO terms according to GO enrichment analysis. For innate immune response related DEGs, significant enrichment was observed in response to virus, response to other organisms, defense response to virus, response to biotic stimulus and regulation of defense response to virus (Figure 3 and Appendix A).

To further define DEG function, KEGG pathway/enrichment analysis was performed. Among the ten significantly enriched pathways, viral carcinogenesis, the RIG-I-like receptor signaling pathway, the cytosolic DNA-sensing pathway and pathways associated with several viral infections (measles, influenza A, Herpes simplex infection, Hepatitis B and C) were found to be enriched to canonical pathways (Figure 4 and Appendix A).

The potential interaction network of the DEGs was examined with STRING analysis. As shown in Figure 5, most of the DEGs related to innate immune response. Among the up-regulated genes, IFIH1(MDA5), IRF7, IRF9, Mx1, RSAD2, IFIT1, IFIT2, OAS1X and OAS1Y were located in the core of the network and linked to many other DGEs. In addition, not all DEGs showed connection with others because their functions were either unrelated or have not yet been clarified. A detailed representation of DEGs is provided in Appendix A.

### 3.4. Protein Profiling of MDBK Cells

After processing the LC-MS/MS spectra using Mascot software, 20,401 unique peptides were mapped to 4501 proteins from MDBK cells, which were assigned to 25 different GO categories. The most abundantly populated GO category, with 588 proteins, was general function (Figure 6 and Appendix A). Other frequently assigned categories included signal transduction mechanisms (526 proteins), post-translational modification, protein turnover, chaperones (473 proteins), intracellular trafficking, secretion, and vesicular transport (357 proteins).

### 3.5. Functional Analysis of Protein Responses to CpHV-1 Infection in MDBK Cells

Based on a cut-off of a fold change (FC) ≥ 1.2 or ≤ 0.83 and a *p* < 0.05, 327 proteins were found to be significantly changed in MDBK cells in response to CpHV-1 infection (Appendix A). This included 147 up-regulated proteins and 180 down-regulated proteins. Among the up-regulated DEPs, ISGs account for 15.65% (23/147). All these ISGs were in the top 85. Among the 23 ISGs, 12 (52.17%) were compatible with the result of RNA-Seq. In addition, the expression of IFIH1 (MDA), DDX58 (RIG-I), IRF7, TRIM5/21/25 and ISG15/20 were also increased. The top five up-regulated DEPs were identified as leucine zipper tumor suppressor 1 (LZTS1), interferon-induced GTP-binding protein Mx1 (Mx1), ubiquitin-like protein ISG15 (ISG15), interferon-induced protein with tetratricopeptide repeats 1 (IFIT1) and OTU deubiquitinase 7A (OTUD7A). The top five down-regulated DEPs were found to be Geranylgeranyl transferase type-1 subunit beta (PGGT1B), PAN2-PAN3 deadenylation complex subunit PAN3 (PAN3), pumilio RNA binding family member 1 (PUM1), CSTF2 protein (CSTF2) and PCNA-associated factor (PCLAF). Taken together, these results indicate that CpHV-1 invasion induced a distinct proteomic profile in MDBK cells, and in turn host cells sharply altered the related proteins in response to CpHV-1 infection.

All DEPs were categorized using GO analysis, based on the international standardized gene functional classification system. They were found to be involved in cellular process (20.08%), biological regulation (14.44%), metabolic process (13.6%) and response to stimulus (11.92%) (Figure 7A). Additionally, some of these proteins were predicted to be associated with cell (18.68%), cell part (18.68%), organelle (16.1%), organelle part (9.66%), membrane (9.18%) and membrane part (6.28%) (Figure 7B). Moreover, proteins were involved in binding (50.79%), catalytic activity (32.46%), transcription regulator activity (4.71%), and molecular function regulator (4.19%) and assigned to transporter activity (3.66%) and structural molecule activity (1.57%) (Figure 7C).

Likewise, the significantly up- and down-regulated DEPs were also annotated with GO analysis. Interestingly, the proteins assigned to biological process (BP), cellular component (CC) and molecular function (MF) for the up-regulated proteins were similar to those assigned to the down-regulated DEPs. The three main BP groups to which proteins identified were assigned were cellular process, biological regulation and metabolic process. Additionally, the CC groups assigned to both the up- and down-regulated proteins were cell, cell part and organelle. However, there are some differences in the MF groups between the up-regulated and down-regulated proteins. The up-regulated proteins were found to be mainly involved in binding, catalytic activity and transcription regulator activity, while down-regulated proteins were found to be related to binding, catalytic activity and molecular function regulators (Figure 8 and Appendix A). GO analysis also showed the differently expressed proteins were enriched in both defense response, response to virus, innate immune response, etc. 

Furthermore, the KEGG pathway analysis for the DEPs (Appendix A) showed that they were involved in the metabolic pathways, Herpes simplex infection, endocytosis, human papillomavirus infection, influenza A, viral carcinogenesis and measles (Figure 9A), which was similar to the results from RNA-Seq. However, a significant difference in the KEGG pathways identified was found between the up-regulated and down-regulated proteins; most of the up-regulated DEPs were associated with the Herpes simplex infection (8.24%, Figure 9B), and the down-regulated DEPs were mainly associated with the metabolic pathways (11.97%, Figure 9C). The pathways associated with viral infection (especially Herpes simplex infection) were enriched under KEGG analysis (Appendix A).

STRING analysis was used to explore the potential interaction network of the DEPs. As shown in Figure 10, most of the DEPs related to innate immune responses. Among the up-regulated genes, DDX58 (RIG-I), IFIH1 (MDA5), Mx1, ISG15, IFIT1, OAS1Y and RASD2 were located in the core of the network and linked to many other DEPs, which was consistent with the RNA-Seq result.

### 3.6. Partial Validation of RNA-Seq and iTRAQ Data

To further validate the RNA-Seq data, eleven selected DEGs (Mx1, RSAD2, IFIT1, IFIT2, IFIT5, IFIH1, IFITM3, IRF7, IRF9, OAS1X and OAS1Y) associated with immune responses from RNA-Seq were selected for qRT-PCR analysis, and they exhibited a high correlation coefficient (R^2^ = 0.87; Appendix A) between RNA-Seq and qRT-PCR results. In addition, qRT-PCR was also performed for samples collected at 24 hpi, and the results also exhibited a high correlation between 12 hpi and 24 hpi (Table 3). Some of the DEGs (DEPs) were further determined by Western blot, as shown in Figure 11; the expression of RSAD2, ISG15 and MX1 were increased significantly, which was consistent with the RNA-Seq or iTRAQ results. These results confirmed that the differential expression genes identified by RNA-Seq and iTRAQ are reliable.

## 4. Discussion

All alpha herpesviruses have some common features, including large double-stranded DNA genome, virion size and structure (icosahedral capsid, tegument and glycosylated lipid envelope), and the latency-reactivation cycle [22]. These viruses are pantropic and neuroinvasive pathogens that establish a persistent or latent infection in the nervous systems of the natural hosts [23]. Upon reactivation, alpha herpesviruses cause diverse effects, which vary from mild epithelial lesions to life-threatening necrotizing brain infections and death [24,25]. For the last 20 years, alpha herpesvirus species, represented by Herpes simplex virus-1 (HSV-1), varicella zoster virus (VZV), pseudorabies virus (PRV) and bovine herpes virus (BHV-1), have been studied to understand the complicated life cycle and pathogenesis [24,26,27,28]. However, current academic research on CpHV-1 focuses on aspects of epidemiological investigation [5,6,7,29] and antiviral treatment [30,31,32]; the pathogenic mechanism of CpHV-1 has been long neglected. The complex and dynamic interplay between virus and host components is an active area of research. The interactions between viral–viral and viral–cell proteins are constantly modified during productive and latent infection. Moreover, the function and localization of viral and cell proteins are regulated at transcriptional, translational and post-translational levels.

As a revolutionary tool, RNA-Seq technology gives the opportunity to produce large numbers of sequence data in non-model organisms [33,34]. The transcriptional and proteomics landscape in the host upon virus infection facilitates the understanding of host immune responses and defense mechanisms based on the pathogenic microorganism infection at the whole mRNA and protein level and provides new approaches to control the viral infections. It has been reported that CpHV-1 replicated efficiently in experimentally infected calves during acute infection and established latent infection, which supports the ability of these viruses to cross-infect the respective heterologous hosts [18]. Moreover, previous studies demonstrated that CpHV-1 could infect goat peripheral blood mononuclear cells, the MDBK cell line and varied human cell lines [14,15,16]. The MDBK cell line used in this study has been widely used in CpHV-1 studies [8,15,30]. A commercial goat cell line was not available until now, and only primary goat testicular/kidney cells were used in our lab. Thus, in the present study, the MDBK cell was selected as a model. It is preferable to study or confirm the cellular responses upon CpHV-1 infection both in bovine and goat cells; we will perform this study in our next-step investigations.

In our study, the transcriptome of CpHV-1-infected MDBK cells were first evaluated at 12 hpi to understand and delineate the mechanism of the early cellular responses induced by virus infection. Transcriptional changes of the DEGs involved in immunological processes were analyzed specially. The results from GO, KEGG and PPI analyses indicated that various numbers of DEGs were involved in different biological processes of host immune responses, especially the innate immunity. iTRAQ in combination with LC-MS/MS analysis has become an important quantitative proteomic method, with certain advantages over traditional proteomic techniques. These advantages include higher throughput, increased sensitivity, and greater accuracy. This technique has been used successfully to explore pathogen–host interactions for both viruses and bacteria. To further validate the RNA-Seq data, CpHV-1-induced modulation of the host cell proteome at 24 hpi (12 h later than RNA-Seq for full translation of different proteins) was analyzed by iTRAQ coupled with LC-MS/MS. The results showed that a total of 327 proteins were significantly changed in MDBK cells after being infected with CpHV-1 for 24 hpi. The appearance of up-regulated DEPs associated with innate immunity (RLRs, ISGs and IRFs) was consistent with those in RNA-Seq. In this study, six well plates were used for virus infection and sample preparation. For each sample used for sequencing, cells from six wells were collected together.

Viral infection constitutes a significant portion of mammalian morbidity and mortality, which has led to extensive investigation into the countermeasures employed by the host to combat such infection. The immune system of organisms acts to eradicate viral infection through disrupting pathways and functions imperative to the pathogen’s life cycle [35]. Innate immunity is the host’s first line of antiviral defense response. The innate immune system has evolved a range of receptors to detect viral pathogens, termed pattern recognition receptors (PRRs), which recognize viral proteins and nucleic acid in the process of non-self-recognition [36]. PRRs (TLRs, RLRs) and associated signaling pathways make up a large part of the innate immune system. Once activated, PRRs initiate a series of signaling cascades that results in the production of the well-known antiviral cytokine, interferon (IFN). IFN is able to act in both an autocrine and paracrine manner to activate the Janus kinase signal transducer and activator of the transcription signaling pathway, resulting in the subsequent downstream expression of hundreds of antiviral host effector proteins, called IFN-stimulated genes (ISGs), which control viral infection in the infected cell and help neighboring cells resist infection [37]. Previous research has elucidated the function of these ISGs and has revealed a broad scope of very specific antiviral mechanisms able to target viruses at varying stages of their life cycle. Some ISGs have a broader spectrum of viral targets, such as radical S-adenosyl methionine domain containing 2 (RSAD2/viperin), MX dynamin like GTPases (Mx1/2), interferon-induced protein with tetratricopeptide repeats (IFITs) and the IFN-induced transmembrane (IFITM) proteins [35].

Miettienen et al. have characterized the secretome of HSV-1-infected human primary macrophages using high-throughput quantitative proteomics. Proteins related to immune and inflammatory responses, IFN-induced proteins, and endogenous danger signal proteins (such as IFIT2, IFIT3, STAT1 and Mx1) were efficiently secreted upon IFN-β priming and HSV-1 infection [26]. In this study, a total of 81 genes were identified as significantly up-regulated, expressed upon CpHV-1-infection. Most of them were mainly involved in the innate immune response, especially the ISGs. These included Mx1, RSAD2, IFITs (IFIT1, IFIT2, IFIT5) and 2’,5’-oligoadenylate synthetase1 (OAS1X and OAS1Y). In proteome analysis, the above genes’ expression was found to be significantly up-regulated. In particular, the expression of ISGs is induced through direct activation of the Toll-like receptor (TLR) or RIG-I receptors by viruses and triggering of the downstream IFN-I signaling pathway [38,39]. In the present study, two RIG-I-like receptors (DDX58 (RIG-I) and IFIH1(MDA5)) and IRF7/9 were found to be significantly up-regulated upon CpHV-1 infection, indicating the activation of associated pathways by CpHV-1.

Mx1 is broadly inhibitory and acts prior to genome replication at an early post-entry step of the virus life cycle. Evidence suggests that Mx1 traps incoming viral components, such as nucleocapsids, and prevents them from reaching their cellular destination [40]. Viperin is an ISG product that plays an important and multifaceted role in the innate immune response to many DNA and RNA viruses [41]. Viperin is expressed at low basal levels in most cell types but is strongly induced by numerous viruses (such as HSV-1, HIV-1, hepatitis C virus and influenza A virus) [42,43,44,45]. Viperin expression is also induced by a wide range of extracellular macromolecules that trigger the innate immune response by engaging various cell-surface receptors; these include type I, II and III interferons, dsDNA and RNA, and lipopolysaccharides [46].

IFIT genes encode a family of proteins that is induced after IFN treatment, viral infection, or pathogen-activated molecular pattern (PAMPS) recognition. IFIT proteins are poised to confer inhibitory effects after infection, and they have been found to be increased in abundance in target cells infected with various viruses, such as infectious bursal disease virus (IBDV) [47], Japanese encephalitis virus (JEV) [48], and porcine reproductive and respiratory syndrome virus (PRRSV) [49]. Recently, progress was made in identifying how IFIT proteins inhibit through distinct mechanisms of action and the replication of multiple families of viruses [50], such as vesicular stomatitis virus [51] and hepatitis C virus [52]. IFIT5 is involved in antiviral responses through enhancing innate immune signaling pathways. Here, IFIT5 was revealed to be increased in abundance during CpHV-1 infection (both transcriptome and proteomic analysis). Therefore, it is conceivable that IFIT5 may be an important modulator in the antiviral innate immune response during CpHV-1 infection.

Furthermore, different from RNA-Seq, protein members in the TRIM (TRIM5, TRIM21 and TRIM25) and ISG (ISG15, ISG20) family were also found to be increased in CpHV-1-infected cells by iTRAQ analysis. The TRIM protein is a novel class of single-protein RING finger E3 ubiquitin ligases [53], characterized by their N-terminal zinc-binding RING and B-box and coiled-coil domain motif, which mediates homomeric and heteromeric interactions among TRIM family members and other proteins [35]. Around one-third of these family members are also ISGs, with many known to be involved in the foundation of an innate antiviral state [54]. TRIM5α, TRIM19, TRIM79α, TRIM56, TRIM21 and TRIM22 have all been demonstrated to specifically inhibit multiple viruses, including a number of flaviviruses, retroviruses, and the hepatitis B virus [55,56]. Many TRIM family members are known to play a role in innate signaling pathways, with a recent screening assay suggesting that as many as half of the family members have a positive role in augmenting innate immune signaling events [35]. The differences of the levels of TRIM and ISG members between RNA-Seq and iTRAQ might result from the different sampling time point and need further examination. Moreover, all sequencing results were based in one biological replicate. In the next step, more replicates will be analyzed, and primary goat testicular/kidney cells will also be inoculated with CpHV-1, and the expression profile of innate immune-related genes will be examined.

Apart from the immune-related DEGs, the expression of Enolase (ENO), was found to be down-regulated. ENO is a multifunctional protein and is involved in many different physiological and pathophysiological processes. The three isoforms of ENOs in mammalian cells include α or non-neuronal enolase (NNE), γ or neuron-specific enolase (NSE) and β or muscle-specific enolase (MSE) [57]. ENO is a key enzyme that catalyzes the interconversion of 2-phosphoglycerate to phosphoenolpyruvate in the glycolytic pathway and gluconeogenesis [58]. It is also responsible for non-glycolytic functions such as contributing regulation of the cytoskeletal filaments [59,60]. Cytoskeletal components are closely connected with viral transport mechanisms within cells, subcellular localization of associated transport systems and maintenance of the viral replication status [59]. Furthermore, viral replication can be effectively suppressed following interference with cytoskeletal components [61,62]. In addition, ENO can regulate virus replication through suppression of the IFN signaling pathway [63]. Notably the proteomic data further revealed that the key glycolytic enzymes, including ENO, were extensively decreased in abundance. Based on these proteomic analyses, we speculate that CpHV-1 replication may extensively inhibit the host cellular metabolic pathways involved in glycolysis and energy metabolism.

To further validate the RNA-Seq data, eleven selected DEGs (Mx1, RSAD2, IFIT1, IFIT2, IFIT5, IFIH1, IFITM3, IRF7, IRF9, OAS1X and OAS1Y) associated with immune responses from RNA-Seq were selected for qRT-PCR analysis, and they exhibited a high correlation coefficient between RNA-Seq and qRT-PCR results. In addition, the selected DEGs also exhibited a high correlation between 12 hpi and 24 hpi. Some of the DEGs (DEPs) were further determined by Western blot. As is shown in Figure 11, the expression of RSAD2, ISG15 and MX1 were increased significantly, which was consistent with the RNA-Seq or iTRAQ results. These results confirmed that the differential expression genes identified by RNA-Seq and iTRAQ are reliable.

Taken together, all the results obtained here indicate that CpHV-1 infection induced the transcription and expression alterations of a series of genes related to host innate immune response and other pathways. These will be useful and helpful for deeply understanding the host–CpHV-1 interaction in the future.

## Figures and Tables

**Figure 1 viruses-13-01293-f001:**
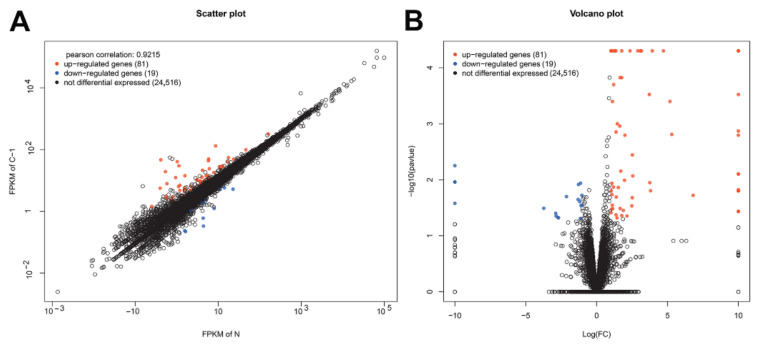
Summary of the differentially expressed genes between Caprine herpesvirus 1 (CpHV-1)-infected and mock-infected samples at 12 hours post infection (hpi). A is a scatter-plot. The abscissa and ordinate coordinates represent the expression of genes or transcripts in two samples, respectively. Pearson correlation refers to the correlation index of gene expression levels of two samples. Red points represent up-regulated genes, blue points represent down-regulated genes, and black circles represent genes showing no significant differences in expression. B is a volcano-plot. The abscissa means the logarithm of the multiple difference of the expression of genes in the two samples; the larger the absolute value, the greater the difference in gene expression. The ordinate means the negative logarithm of the statistical significance of the expression of gene; the larger the value, the more significant the differential expression and the more reliable the differential gene. Red points represent up-regulated genes, blue points represent down-regulated genes, and black circles represent genes showing no significant differences in expression.

**Figure 2 viruses-13-01293-f002:**
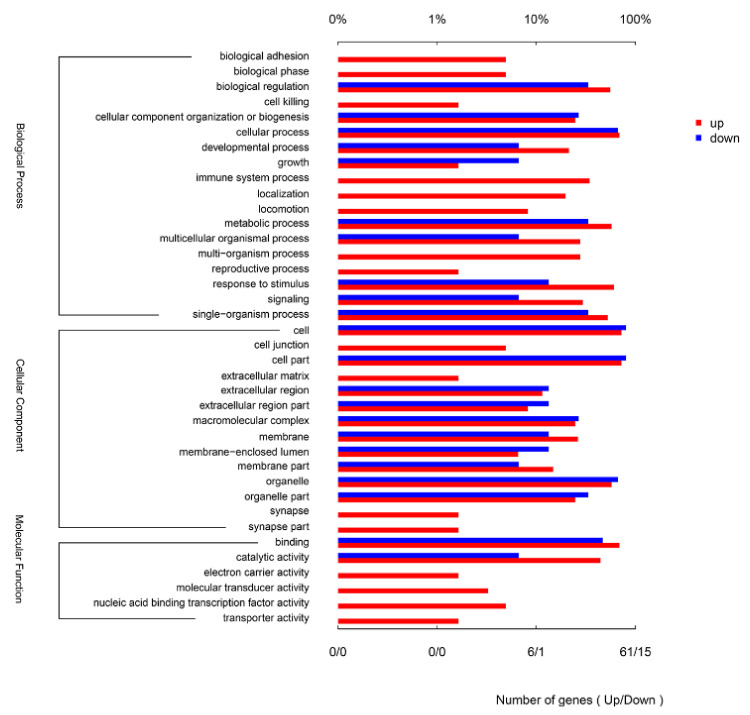
Gene Ontology (GO) annotation for the differentially expression genes (DEGs) between CpHV-1-infected and mock-infected Madin Darby bovine kidney (MDBK) cells. DEGs were annotated in three categories: biological processes, cellular components and molecular functions. The upper abscissa represents the proportion of the percentage of the number of genes corresponding to the function, and the lower abscissa represents the number of genes corresponding to the function. Red columns represent up-regulated genes, blue columns represent down-regulated genes.

**Figure 3 viruses-13-01293-f003:**
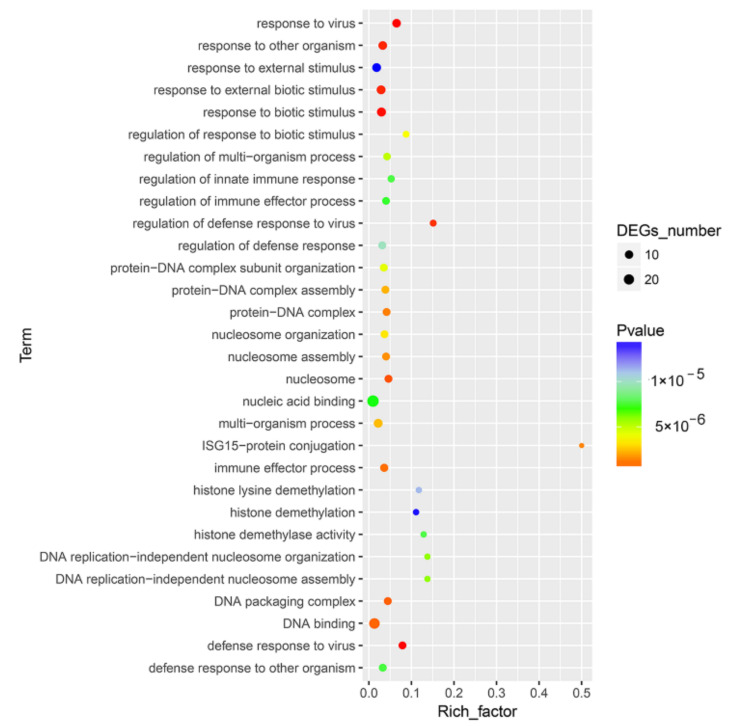
GO enrichment analysis for the DEGs between CpHV-1-infected and mock-infected MDBK cells. The abscissa represents the enrichment factor, and the ordinate represents the GO function classification. Circles indicate numbers of enriched genes, and colors depict the *p*-value. The enrichment factor was calculated using the number of enriched DEGs divided by the total number of background genes in the corresponding pathway. The size of each circle represents the number of significant DEGs enriched in the corresponding pathway. The chromatogram from blue to red represents the uncorrected *p*-value. A detailed representation of the data is provided in Appendix A.

**Figure 4 viruses-13-01293-f004:**
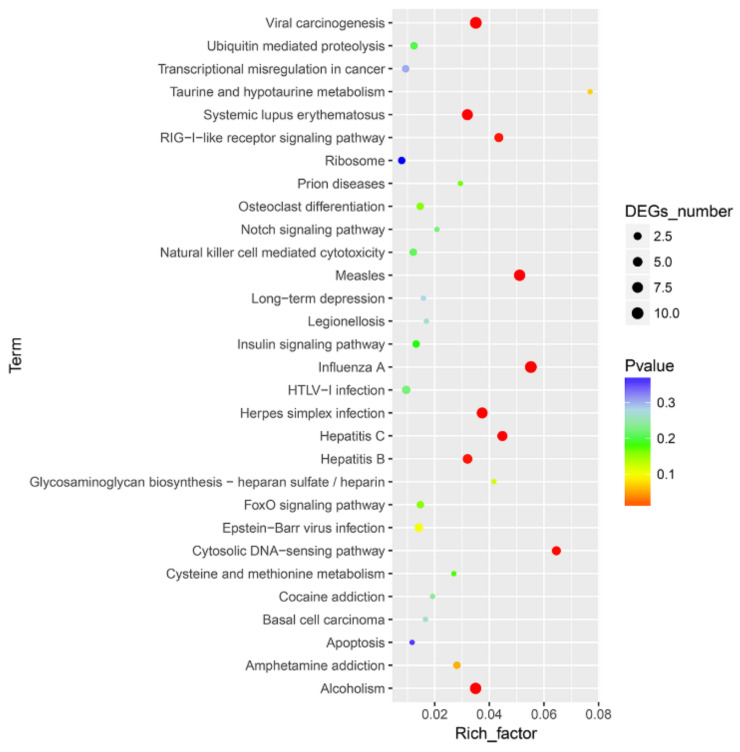
Top 30 pathways enriched in Kyoto Encyclopedia of Genes and Genomes (KEGG) database for the DEGs between CpHV-1-infected and mock-infected MDBK cells. The abscissa represents the enrichment factor, and the ordinate represents the classification of the KEGG metabolic pathway. Circles indicate numbers of enriched genes, and colors depict the *p*-value. The size of each circle represents the number of significant DEGs enriched in the corresponding pathway. The enrichment factor was calculated using the number of enriched DEGs divided by the total number of background genes in the corresponding pathway. The chromatogram from blue to red represents the uncorrected *p*-value. A pathway with a *p*-value < 0.05 is considered to be significantly enriched. A detailed representation of the data is provided in Appendix A.

**Figure 5 viruses-13-01293-f005:**
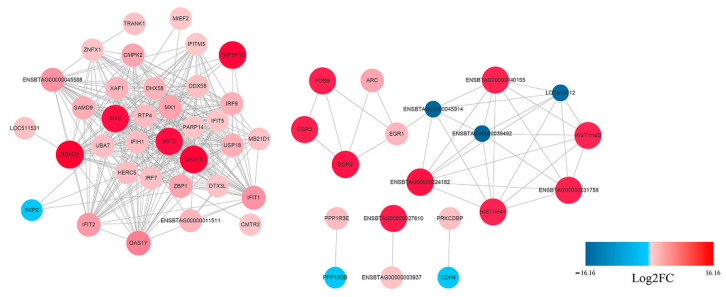
Protein–Protein Interaction (PPI) network of the selected DEGs based on STRING analysis results and fold change information. The network nodes represent the proteins encoded by the DEGs. The up-regulated genes are shown in red, and the down-regulated genes are shown in blue, with the gradient showing the extent of expression. The size of the node indicates connectivity. A detailed representation of DEGs is provided in Appendix A.

**Figure 6 viruses-13-01293-f006:**
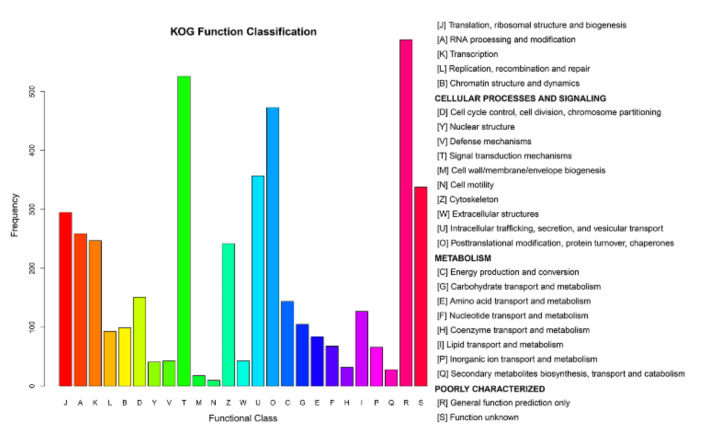
Clusters of Orthologous Groups (COG) annotations of the proteome of MDBK cells infected with CpHV-1. The abscissa represents the function code of the COG database. The description information of the function code is shown on the right side of figure. The ordinate represents the abundance value of each function code. A detailed representation of the data is provided in Appendix A.

**Figure 7 viruses-13-01293-f007:**
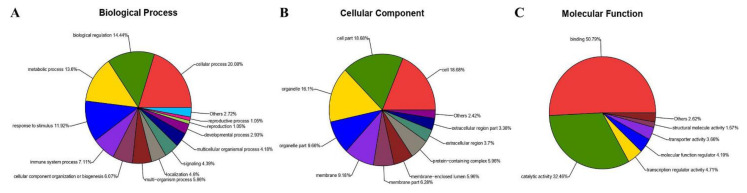
Gene ontology analysis of 327 proteins differentially expressed in MDBK cells infected with CpHV-1. Proteins were annotated by biological process (**A**), cellular component (**B**), and molecular function (**C**). Each color represents a different GO term, and the percentages of protein numbers in each category are shown after the name of the GO term. A detailed representation of the data is provided in Appendix A.

**Figure 8 viruses-13-01293-f008:**
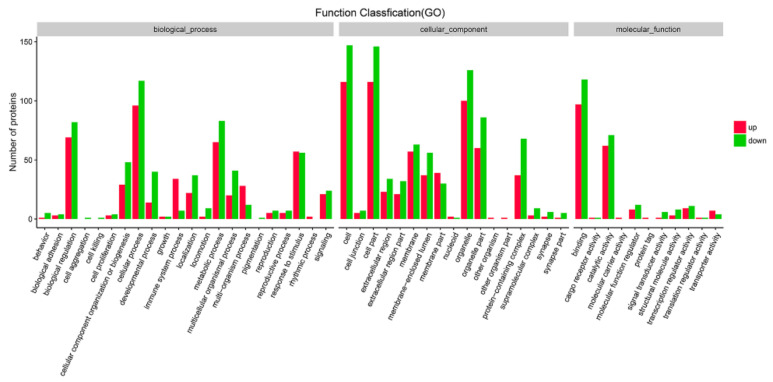
Gene ontology classification of up- and down- regulated proteins in MDBK cells infected with CpHV-1. The differentially expressed proteins (DEPs) were annotated in three categories: biological processes, cellular components and molecular functions. The abscissa represents the GO function classification, and the ordinate represents the number of proteins corresponding to the function. Red bars indicate upregulated proteins, and green bars indicate down-regulated proteins. A detailed representation of the data is provided in Appendix A.

**Figure 9 viruses-13-01293-f009:**
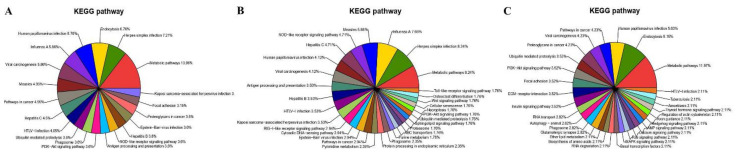
KEGG pathway classification of DEPs in MDBK cells infected with CpHV-1 (**A**), including up-regulated proteins (**B**) and down-regulated proteins (**C**). Each color represents a different KEGG pathway, and the percentages of enriched target protein numbers in each category are displayed in brackets. A detailed representation of the data is provided in Appendix A.

**Figure 10 viruses-13-01293-f010:**
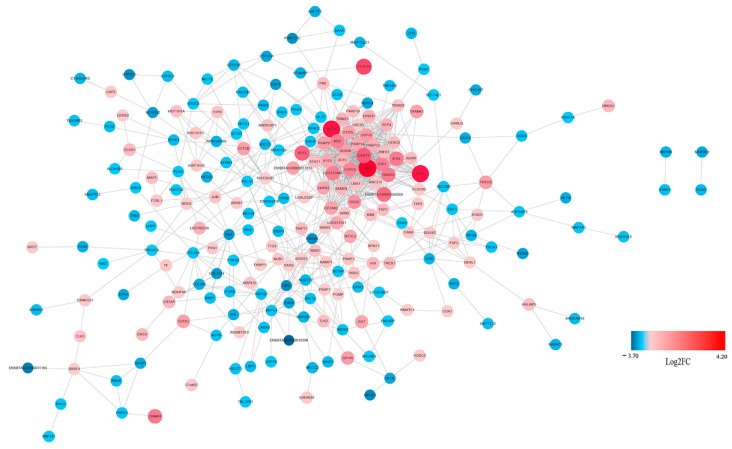
PPI network of the selected DEPs based on STRING analysis results and fold change information. The network nodes represent DEPs. The up-regulated proteins are shown in red, and the down-regulated proteins are shown in blue, with the gradient showing the extent of expression. The size of the node indicates connectivity. A detailed representation of DEPs is provided in Appendix A.

**Figure 11 viruses-13-01293-f011:**
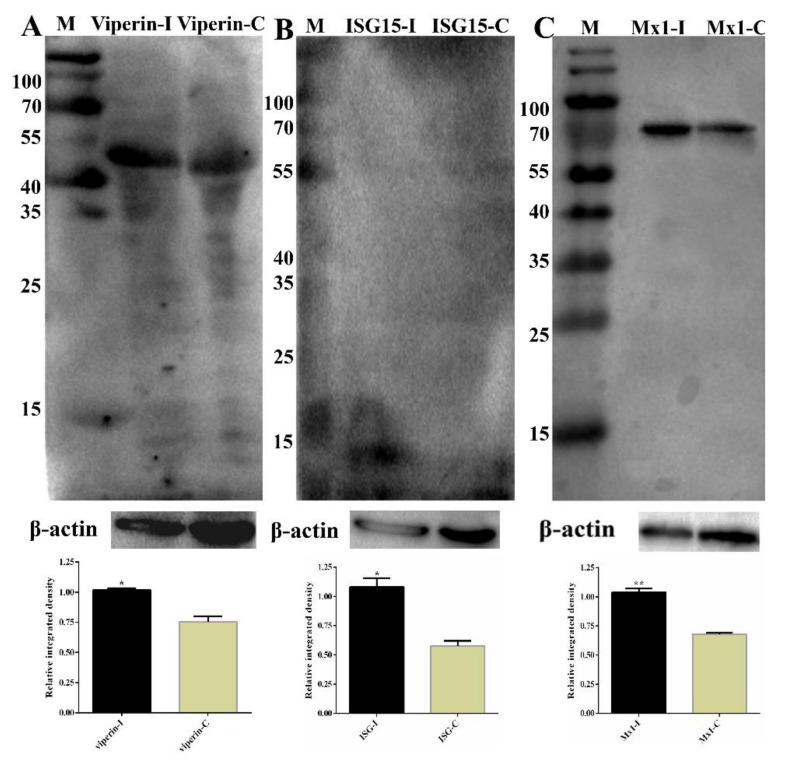
Partial validation of RNA-Sequencing (RNA-Seq) and isobaric tags for relative and absolute quantitation (iTRAQ) data by Western blot. (**A**) The expression of RSAD2 (Viperin) in the samples was determined by Western blot. The molecular weight of Viperin was 42 kDa. Lane M, prestained protein mass markers; Lane 1, CpHV-1-infected MDBK cell sample; Lane 2, mock-infected MDBK cell sample. β-actin was the internal control; molecular weight was 42 kDa. (**B**) ISG15 western blot analysis of MDBK cells. The molecular weight of ISG15 was 15 kDa. Lane M, prestained protein mass markers; Lane 1, CpHV-1-infected MDBK cell sample; Lane 2, mock-infected MDBK cell sample. β-actin was the internal control. (**C**) MX1 western blot analysis of MDBK cells. The molecular weight of MX1 was 74.8 kDa. Lane M, prestained protein mass markers; Lane 1, CpHV-1-infected MDBK cell sample; Lane 2, mock-infected MDBK cell sample. β-actin was the internal control. The integrated density of every group was compared with that of internal loading control β-actin before the relative integrated density values between every infected group and its corresponding control group were calculated. Western strips were performed by grayscale analysis using the ImageJ software, and error bars represent relative integrated density mean standard deviations among three independent replicates. * *p* < 0.05 and ** *p* < 0.01.

**Table 1 viruses-13-01293-t001:** Primers used for quantitative real-time RT-PCR (qRT-PCR).

Gene	Accession No.	Primer	Target Position	Product Length (bp)
IRF7	NM_001105040.1	F: 5′-GCTCCACTACACCGAGAAGC-3′	1367-1386	196
		R: 5′-GAAGTCAAAGATGGGCGTGT-3′	1562-1543	
IRF9	NM_001024506.1	F: 5′-AAGGCCTGGGCGATATACAA-3′	694-713	125
		R: 5′-CCGATCTCAGGAACCTCCTC-3′	818-799	
IFIT1	XM_015469501.1	F: 5′-GGAACGTGCTGTGCAACTAA-3′	1305-1324	134
		R: 5′-TGTCGAGTGCTTTCATGCAG-3′	1438-1419	
IFIT2	XM_001787823.5	F: 5′-GGCAGCAAAGCTGTATCGAA-3′	975-994	137
		R: 5′-CTTCCAGGACTTTGGCCCTA-3′	1111-1092	
IFIT5	NM_001075698.1	F: 5′-CGTGGAGCGAGACTCTATGT-3′	1160-1179	162
		R: 5′-CGGCCATAGTGGTAGTGGAT-3′	1321-1302	
IFIH1	XM_010802053.1	F: 5′-ATTCTGAGGCAGACGGGAAA-3′	896-915	74
		R: 5′-TCTGTACTGCCTTCACAGCA-3′	969-950	
IFITM3	NM_001078141.2	F: 5′-TGGTCCCTGTTCAACACCAT-3′	231-250	91
		R: 5′-CCATCTTCCGGTCCCTAGAC-3′	321-302	
OAS1X	NM_178108.2	F: 5′-AGCACTGGTACCAACTGTGT-3′	671-690	132
		R: 5′-GAAATCCCTGAGCTGTGCTG-3′	802-783	
OASIY	NM_001040606.1	F: 5′-CTCAGCTTTGTGCTGAGGTC-3′	453-472	131
		R: 5′-TGGATGAGCCGGACATAGAC-3′	583-564	
MX1	JQ766265.1	F: 5′-TTTTTCAACCTCCACCGAAC-3′	1450-1469	132
		R: 5′-GTACACCTGGTCCTGGCAGT-3′	1581-1562	
RSAD2	NM_001045941.1	F: 5′-TTCAACGTGGACGAGGATATG-3′	734-754	98
		R: 5′-CCAGAGTTCTCACCCTCAATTAT-3′	831-809	
β-actin	AY141970.1	F: 5′-ACATCCGCAAGGACCTCTA-3′	902-920	89
		R: 5′-GATCTCTTTCTGCATCCTGTCC-3′	990-969	

**Table 2 viruses-13-01293-t002:** Summary of read quality and mapping results of RNA-Sequencing (RNA-Seq).

Sample	Total_Raw Reads	Total_Clean Reads	Mapped_Reads	Mapped_Rate	FPKM ^a^ > 0	FPKM > 1	FPKM > 5	FPKM > 10
C-1	30,065,892	25,480,936	19,704,426	77.33%	13,345	10,888	8437	6426
N	26,223,456	22,774,082	17,675,904	77.61%	13,153	10,864	8426	6431

a: FPKM: Fragments Per Kilobase of exon model per Million mapped reads.

**Table 3 viruses-13-01293-t003:** Up-regulated targets in this study.

Gene/Protein Name	Fold Change (FC) (CpHV-1-Infected/Mock-Infected)			
	RNA-Seq (log2)	qRT-PCR (12 hpi, log2)	qRT-PCR (24 hpi, log2)	iTRAQ
MX1	3.93	6.86	6.6	4.20
MX2	/ ^a^	/	/	2.35
RSAD2 (Viperin)	inf ^b^	10.59	13.06	2.42
IFIT1	5.3	4.85	7.67	3.80
IFIT2	4.71	5.34	5.16	1.76
IFIT3	/	/	/	2.78
IFIT5	1.09	1.35	2.42	1.5
IFITM3	1.008	0.61	1.6	/
IFI44	/	/	/	2.43
OAS1X	inf	8.87	8.93	1.76
OAS1Y	6.81	3.21	4.39	2.86
IRF7	1.63	0.48	0.36	1.41
IRF9	2.91	3	2.76	/
IFIH1 (MDA5)	1.77	4.1	3.11	2.34
DDX58 (RIG-I)	/	/	/	1.83
TRIM5	/	/	/	1.31
TRIM21	/	/	/	1.60
TRIM25	/	/	/	1.32
ISG15	/	/	/	3.81
ISG20	/	/	/	2.05

a: no significant difference was found or not tested by qRT-PCR; b: infinity.

## Data Availability

All the data we used and analyzed in the current study are available in the manuscript and Appendix A. RNA-Sequencing data were submitted to NCBI, with the SRA accession number PRJNA728547. The mass spectrometry proteomics data have been deposited in the ProteomeXchange Consortium via the PRIDE [64] partner repository, with the dataset identifier PXD026014.

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
