# Peer review of "Transcriptome and Proteomic Analysis Reveals Up-Regulation of Innate Immunity-Related Genes Expression in Caprine Herpesvirus 1 Infected Madin Darby Bovine Kidney Cells"

_viruses, 2021, doi:10.3390/v13071293_

Round 1

Reviewer 1 Report

Dear Editor,

The manuscript entitled “Transcriptome and Proteomic Analysis Reveals Up-Regulation of Innate Immunity Related Genes Expression in Caprine Herpesvirus 1 Infected Madin Darby Bovine Kidney Cells” by Fei Hao et al. presents the transcriptomic and proteomic analysis of CpHV-1 infected Madin Darby bovine kidney (MDBK) cells by RNA-Seq and iTRAQ-LC-MS/MS technology, respectively. RNA-Seq analysis revealed 81 up-regulated and 19 down-regulated differentially expressed genes between infected and mock-infected MDBK cells, mostly involved in the innate immune response (interferon sitimulated genes (ISGs)). On the other hand, proteomic analysis showed significant up-regulation of innate immunity related proteins and protein-protein interaction network analysis indicated most of the DEGs related to innate immune responses. The authors claim that their findings support the notion that CpHV-1 infection induced the transcription and protein expression alterations of a series of genes related to host innate immune response, which helps to elucidate the resistance of host cells to viral infection and to clarify the pathogenesis of CpHV-1.

In my opinion, the manuscripts’ findings are interesting, the manuscript is well-written and should be accepted for publication after major revisions. My detailed comments for the authors to consider are provided below:

  1. The authors should explain in a more detailed way why they chose a bovine cell line to study the caprine virus. They state that the cell line has previously used in several studies however they should state what are the advantages and the drawbacks of using that specific cell line for studying CpHV-1.
  2. The authors state that they use a 12 hpi period for RNA-Seq analysis and 24 hpi for proteomic analysis. I would like to see a wider discussion with specific references for reasoning these time points selection.
  3. My main concern is that I cannot see more than one biological replicate in the analysis. That fact brings a certain degree of uncertainty on the results. Ideally, more replicates should have been analyzed. In case that is not possible, the present findings are interesting but it should be pointed throughout the discussion section that all conclusions are based in one replicate and thus further study is needed.
  4. A reference is needed for Cytoscape.
  5. Paragraph 2.8 should be re-written in passive voice.
  6. In page 6, line 244, the authors write that the reference genome belongs to Gallus gallus. The link refers to Capra hircus which is the one used I suppose. Please correct that. Please explain why a goat reference genome was used instead of a bovine related genome since the cell line used is bovine.
  7. Figure S1 does not exist.
  8. Figures 5,7 and 9 are not readable, even in high magnification. Please provide them in better resolution.
  9. Figures 11A and 11B should be replaced. They are overexposed and maybe misleading.
  10. I would like to see the RNA-Seq – RT-PCR correlation figure, maybe in the supplementary material.

Reviewer 2 Report

The paper by Hao et al is a detailed account of the transcriptional and proteomic landscape of host MDBK cells (bovine) infected with Caprine herpesvirus 1.  The paper uses both transcriptomic analysis of infected cells, paired with proteomic analysis using iTRAQ and LC-MS/MS to try and link gene expression and functional studies.  Analysis of functional and protein-protein interactions was performed in addition to KEGG pathway/enrichment analysis and protein profiling.     The authors identified 81 genes being upregulated and 327 proteins with significantly changes in MDBK cells after 24 hours, with the majority of these being involved in innate immunity and being consistent with RNA-Seq.  Some results were also partially validated using quantitative real-time PCR and western blotting.

In my opinion, whilst the overall novelty of the work is limited mainly to that of Caprine Herpesviruses, it will make contributions to research in the wider field of Herpes virus virology.  Its main strength being the multi-discipline analysis of genomic and proteomic investigation.  Overall, it is well written and of sound experimental design, although I do have some minor critique for the authors.  Overall, I feel this paper is of sufficient quality to publish.

Minor:

  • How many replicates were pooled to make the samples (line 88-89)?
  • What was the specific library prep kit used to prepare the libraries (line 98)?
  • Could the authors expand a little on why Beta-actin selected as a single house-keeping gene – was there supporting data to show this (line 146)?
  • the following sentence requires some rewriting for clarity (line 49): Evidences proved that CpHV-1 causes systemic symptoms in young kids.....
  • Sentence line 409-411 - "upon"repeated twice in same line.
  • On some of the figures, the text is too small to read.  Would it help to put some in supplementary data?

Round 2

Reviewer 1 Report

I would like to thank the authors for revising the manuscript according to my  comments. They have addressed all issues raised.
Therefore, I recommend publication in the present revised version.